# Reaching Natural Growth: Light Quality Effects on Plant Performance in Indoor Growth Facilities

**DOI:** 10.3390/plants9101273

**Published:** 2020-09-27

**Authors:** Camilo Chiang, Daniel Bånkestad, Günter Hoch

**Affiliations:** 1Department of Environmental Sciences—Botany, University of Basel, Schönbeinstrasse 6, 4056 Basel, Switzerland; guenter.hoch@unibas.ch; 2Department of Research and Development, Heliospectra, Fiskhamnsgatan 2, 414 58 Gothenburg, Sweden; daniel.bankestad@heliospectra.com

**Keywords:** light quality, blue light, red light, LED, phytotrons, photosynthesis, photomorphology

## Abstract

To transfer experimental findings in plant research to natural ecosystems it is imperative to reach near to natural-like plant performance. Previous studies propose differences in temperature and light quantity as main sources of deviations between indoor and outdoor plant growth. With increasing implementation of light emitting diodes (LED) in plant growth facilities, light quality is yet another factor that can be optimised to prevent unnatural plant performance. We investigated the effects of different wavelength combinations in phytotrons (i.e., indoor growth chambers) on plant growth and physiology in seven different plant species from different plant functional types (herbs, grasses and trees). The results from these experiments were compared against a previous field trial with the same set of species. While different proportions of blue (B) and red (R) light were applied in the phytotrons, the mean environmental conditions (photoperiod, total radiation, red to far red ratio and day/night temperature and air humidity) from the field trial were used in the phytotrons in order to assess which wavelength combinations result in the most natural-like plant performance. Different plant traits and physiological parameters, including biomass productivity, specific leaf area (SLA), leaf pigmentation, photosynthesis under a standardised light, and the respective growing light and chlorophyll fluorescence, were measured at the end of each treatment. The exposure to different B percentages induced species-specific dose response reactions for most of the analysed parameters. Compared with intermediate B light treatments (25 and/or 35% B light), extreme R or B light enriched treatments (6% and 62% of B respectively) significantly affected the height, biomass, biomass allocation, chlorophyll content, and photosynthesis parameters, differently among species. Principal component analyses (PCA) confirmed that 6% and 62% B light quality combinations induce more extreme plant performance in most cases, indicating that light quality needs to be adjusted to mitigate unnatural plant responses under indoor conditions.

## 1. Introduction

Temperature and light are principal determinants of plant growth, as plants react to environmental conditions in their development. With improvements in controlled environment facilities, the use of indoor cultivation systems has increased worldwide, both for research and plant production. One of the problems, that especially plant researchers are confronted with, is a clear difference between plants grown under indoor versus outdoor conditions. These differences are limiting the transferability of results from indoor experiments to natural systems. Several experiments have tried to replicate outdoor growth in indoor facilities, but low correlations have been found [1,2]. Poorter et al., [3] suggested that this difference comes mainly from the different photothermal ratio (PTR), the ratio between the daily light integral and the daily mean temperature, which is generally much lower in growth chambers. The low PTR in indoor experiments mainly derives from the low and constant irradiances used, compared with the higher and variable sunlight conditions found in nature. In general, conditions in indoor facilities lead to higher specific leaf area (SLA), leaf nitrogen content, and relative growth rate. While maximum photosynthesis (A_max_), plant height, and shoot dry weight (SDW), are lower compared with outdoor experiments [3].

Due to the high photosynthetic efficiency of blue (B) and red (R) light, high electrical efficiency of B and R LEDs, as well as the high technical requirements to create sun-like LED spectra [4,5], most existing indoor plant growth facilities with LED lighting systems use mixtures of mainly B and R light. However, different LED lamps use different proportions of B and R LEDs, or B and R in combination with other LED types, such as white and far-red. This results in very different lighting environments among different indoor growth facilities. In addition, the lack of a common protocol for reporting and measuring LED light irradiance further limits the comparability between experiments [6]. Many studies have investigated plant response to different B to R ratios. These studies revealed that independent of light intensity, a required minimum percentage of B light is necessary to maintain the activities of photosystem II and I [7]. Hogewoning et al., [8] suggested that at least 7% B light is necessary to reproduce near-natural plant growth. In addition, it has been observed that long exposures of monochromatic light can have drastic effects, including non-natural morphologies. With parameters such as shoot elongation, specific leaf area (SLA), chlorophyll concentration and photosynthetic performance being affected [9,10,11,12].

The vast majority of studies related to light quality effects on plants have been conducted under low light levels, varying between 20 to 330 µmol m^−2^ s^−1^ [13,14,15,16,17,18], with a few exceptions (for example 550 µmol m^−2^ s^−1^ [19]), even though interactions between light quantity and quality have been reported previously [9]. Finally, it is also important to consider other light quality related parameters, for example, the effect of red to far red ratio (R:FR). The applied light conditions in indoor cultivation typically has a much higher R:FR ratio (or a complete absence of FR) compared with sunlight conditions. This affects plant photosynthesis, morphology, and development (for example [8,10,14,15,18,19,20]). Once the R:FR ratio is corrected to more natural values, a more natural-like growth may be achieved, despite the large deviations from natural sunlight in other parts of plant biologically active radiation (280–800 nm; for example [21])

The aim of this study is to provide the first step in a series of experiments with the overall goal of reaching nature-like growth of plants under indoor conditions. Specifically, we investigate the effects of varying proportions of B and R light within walk-in growth chambers (phytotrons) on growth and physiological traits of plants from different functional groups. We also compared our findings to the same species grown in a natural-light field trial, where we expected more “natural-like” growth in our indoor treatments that applied a closer to natural light spectra. The inclusion of seven different species from different functional plant types further enabled us to identify if light quality affects plant performance differently among species and plant types. In contrast to many previous studies, we explicitly applied more natural-like R:FR ratios and light intensities [8,9,10,11,12,13,14], and the plants were exposed to temperatures and air humidity based on the pre-measured field trial.

## 2. Results

### 2.1. Light Treatments

Four different treatments were obtained through calibrating the phytotrons for the desired spectra as indicated in Table 1 and Figure 1.

### 2.2. Plant Growth and Biomass Allocation

There was a significant interaction between the light treatments and the different species on the total plant height at the end of the experiments (Table 2), where the relationship with the field trial was species dependent. Some species, for example, *Alnus* and *Melissa*, were significantly smaller independent of the light treatment, while others, for example, *Ocimum*, were taller than the same species in the field trial.

Comparing only among the phytotron treatments, all species had shorter individuals at higher percentages of blue (B) light (62%), which was most pronounced in *Alnus* and *Melissa* (58 and 52% lower height respectively, compared with the 6% B treatment; Figure 2A). Other species like *Ocimum* and *Triticum* were less affected by changes in B light, but follow the same trend (20 and 15% lower height respectively, compared with the 6% B treatment; Figure 2A). In several of the tested species, there was a significant difference in plant height between the two intermediate B treatments (25 and 35% B). Averaged across species, 6% B light produced 22% taller plants that were statistically significantly different from the two intermediate treatments. While in the other extreme, 62% B light yielded a statistically significant shortening of plants by approximately 20% compared with the average across treatments (Figure 2A). A dose response was obtained for specific leaf area in several species (SLA, Figure 2B). Unlike the height results, and due to the species-specific reactions to the light treatments, the average response across species did not significantly differ, neither within the light treatments, nor between the light treatments and the outdoor control. However, *Lactuca* and *Alnus,* for example, had significant higher SLA at 6% B compared with other light treatments, while other species, for example, *Raphanus* and *Triticum,* had higher values at 25 or 35% B light compared with 6 or 62% B light.

There were significant interactions between the light treatments and species for the dry biomass of leaves, shoots, roots and the total dry biomass (Table 2). Similar to plant height and SLA, the relationship between plant biomass and light, under the different light treatments, with the field control was species dependent, yet averaged across all species. Leaf biomass did not significantly differ from the outdoor control in any of the light treatments.

If only the phytotron treatments are compared, there was a lower leaf biomass under 62% B light compared with 6% B light in all investigated species. This was especially the case for the two tree species tested, where *Alnus* and *Ulmus* were most sensitive to high percentages of B light (Figure 3A). On average, plants exposed to 6% B had 35% higher leaf biomass than plants exposed to 62% B (Figure 3A). Similar results were obtained for shoot biomass where, across all species, plants grown at 62% B had a significantly lower shoot biomass compared with all the other light treatments, and yet similar values as in the field trial (except for *Ulmus* and *Ocimum*, Figure 3B). In contrast to the aboveground biomass, the effects of light quality on root biomass were different among all species (Figure 3C). In comparison to the field trial, four species (*Ulmus*, *Lactuca*, *Ocimum*, *Triticum*) had significantly higher root biomass in the phytotron treatments, while in three species (*Raphanus*, *Alnus*, *Melissa*) it was similar compared to the field trial (Figure 3C).

Across all species, there was no strong effect of light quality on root biomass, but a trend to higher root biomass at 6% B (Figure 3C). Total biomass production followed the same trend as found for the individual plant organs, with a significant interaction between light treatment and species (Table 2); higher values under indoor conditions independent of the light treatment, compared to the field trial and increasing biomass with increasing percentage of blue light (data not shown).

With respect to the effect of light quality on the allocation of biomass, there was a significant interaction between light treatment and species for the root to shoot (r:s) mass ratio (Table 2). Almost all species had significantly higher r:s values in the phytotrons compared to the field trial independent of the light treatment, with Triticum showing a four to eight times higher investment in roots compared with the field control (Figure 3D). In some species (e.g., *Alnus* and *Ocimum*), 6% and 62% B light induced higher r:s ratios than 25 and 35% B light, while other species (e.g., *Melissa* and *Ulmus*) were almost indifferent with respect to light quality (Figure 3D).

### 2.3. Leaf Pigmentation

There were significant interactions between the different treatments and species in the pigment concentrations of the leaves (Table 2). Furthermore, the difference between the field trial and the different light treatments was species dependent, but all investigated species exhibited higher Chl a concentration in leaves at 62% B light compared to the other light treatments (strongest effect in *Lactuca*) and several species exhibited the lowest Chl a concentrations at 6% B light (Figure 4A).

On average across all species, 6% B was the only treatment significantly different from the field trial, with 24% lower concentration of Chl a. The effect on Chl b was similar to that of Chl a, with a smaller effect of the light quality on the total amount of Chl b (data not shown). As a result, the average a:b ratio across all species was not significantly different among the light treatments, but significantly higher than in the field trial (Table 2, Figure 4B). The concentrations of carotenoids in leaves, showed overall very similar reactions to light quality as chlorophyll, with increasing concentrations at higher proportions of blue, and an interaction between the light treatment and species (Figure 4C, Table 2). Like chlorophyll and carotenoids, the Fv/Fm values, showed significant interaction between the species and the light treatments (Table 2). Almost all species in the phytotron treatments with 25, 35 and 62% B had Fv/Fm values close to the field trial (Figure 4D), except *Ocimum*, which revealed higher Fv/Fm values indoors than in the field. Averaged across all species, Fv/Fm was significantly lower than in the field at 6% B (Figure 4D). Performance index (Pi) absolute values followed the same trend as Fv/Fm (data not shown, Appendix A).

### 2.4. Photosynthesis and Leaf Respiration

In contrast to the other plant traits tested, all species reacted uniformly to the light treatments in all measured photosynthesis and leaf gas exchange parameters, with no significant interaction between treatment and species effect found (Table 2). When measured with the standardised light of the gas exchange chamber, the average maximum photosynthesis (A_max_) across all species was significantly higher in plants raised at 62% B compared with the field trial (Figure 5A). Meanwhile, when the same parameter was measured under the in situ light, higher values were reached at either 25% or 35% B light compared with the field trial (Figure 5B). The quantum yield of the CO_2_ fixation (α) had similar trends to A_max_, where on average no light treatment was significantly higher than the field trial when the standardised light was used. The 62% B light was the only treatment to induce higher α values than the other light treatments (Figure 5C). When α was measured using the in situ light, higher values were reached at either 6%, 25% or 35% B compared to the field trial (Figure 5D).

The photosynthetic light compensation point (CP) and the dark respiration of leaves (DR) were significantly different among species (Table 2). Averaged across all species, there were no significant effects of the treatments on CP when the standardised light was used. However, with in situ light significantly lower values were reached under 6 and 25% B conditions, compared with 35 and 62% B and the field trial (data not shown). DR was on average significantly lower in plants exposed to 62% B light compared with other light treatments and the field trial when the standardised light was used (Figure 5E). This was not the case for the in situ light, where although several species had higher DR values than the field trial, no significant difference was found between the treatments for the average across species (Figure 5F).

### 2.5. Principal Component Analysis (PCA)

Principal component analysis (PCA) for each species revealed a clustering of each treatment with varying degrees of overlap (Figure 6); from easily differentiable groups between light treatments in some species, for example, *Alnus*, *Lactuca* and *Triticum*, to a more continuous gradient among treatments.

*Melissa*, *Raphanus*, *Alnus*, *Ocimum*, *Lactuca*, and *Triticum* showed a large variability between treatments from outdoor (field trial) to indoor conditions, while the different light treatments tended to cluster. This was not the case for *Melissa*, *Raphanus*, and *Ulmus*, where the field trial was not clearly separated from the phytotron treatments (Figure 6). The two intermediate treatments (25% and 35% B) yielded responses closer to the average (i.e., the centre of the figure) in most species. The loadings for score calculations were also plotted to determine the importance of each factor. No single parameter was specifically responsible for the variation across treatments and between species, except for CP in *Ocimum* growing in the field trial (Appendix A). Independent of the species the first two components explained between 31% and 43% of the total variability.

## 3. Discussion

Previous studies investigating the effect of the spectral light quality on plant performance were mainly focused on single species, and they generally did not directly compare findings with natural conditions. In the present study, we deliberately investigated a suite of species from different functional plant types to determine if, and how, they react to the different treatments. Through application of the same mean climatic conditions indoors, as in the initial field trial, we could better assess which LED light conditions are generating the most natural-like plant performance. Our results showed clear differences within and between the light treatments when compared to the field trial on most measured plant traits. The effect sizes were highly species-specific, while effect directions were similar among species, with the clear exception of SLA and root biomass production. As expected, light treatments with very extreme blue: red (B:R) ratios (6 and 62% B) induced more extreme (‘unnatural’) values in most plant traits than treatments with a more balanced B:R ratio (25 and 35% B).

### 3.1. Light Quality Effects on Morphology

Studies that compared indoor with outdoor plant growth were previously often biased by a higher plant density in the indoor condition [3]. In our study, we deliberately kept the exact same plant densities between the field and the phytotron trials to avoid any stand density bias on plant morphology. The effects of B light percentages on plant morphology have been previously reported in several studies [8,11,12,21,22,23,24,25]. In general, B light is sensed by the cryptochrome system, where under high irradiances or high levels of B light, plants exhibit shorter and stunted growth (For example [8,14,26]). It is also known that a total lack of B or R light negatively affects plant performance, including growth rate, height, photosynthesis and several other parameters. For example, Hernandez et al. [10] found that tomato plants grew shorter under either B or R light mixtures compared with only B or R light.

Previous studies have shown that under high levels of B light, there is an increase in the palisade cell area, which can lead to an increase in leaf thickness (For example [8,10,12]). However, this B light-induced increase in leaf thickness does not necessarily have to translate into a lower SLA [27]. Dougher and Budgee [22] identified that the direction of the effect of B light on SLA is very species dependent. Independent of the applied light quality, Poorter et al. [3] found that on average, indoor experiments tend to produce plants with higher SLA compared to field grown plants, mainly due to higher temperatures and lower light quantity in indoor facilities. In our study, which applied the average temperature and light quantity as in the field trial, the SLA of most species was similar between plants growing in the phytotrons and in the field.

Under the different treatments stem, leaf, root, and total dry biomass largely followed the trend in plant height. The lower biomass at high B% can thus be explained by a stronger inhibition of stem elongation by B light due to an increased cryptochrome activity [14], exposing the plants to lower irradiance due to larger distances to the light source compared with plants treated under a lower percentage of B light. In addition, the stunted growth of plants at high B% leads to an increased self-shading of leaves and decrease in light interception, which has been proposed to result in negative consequences for the whole plant productivity [21]. Although the individual species reacted differently between phytotrons and the field trial, on average, a significantly higher plant biomass within our phytotron treatments compared with the field was found (except for the 62% B treatment). In contrast, Poorter et al. [3] reported lower biomass under indoor conditions compared with field grown plants depending on species and functional group. Again, this apparent contradiction could be explained by the fact that in contrast to other indoor experiments, we deliberately applied the same average temperatures and light strength in the phytotrons as were measured in the field trial. Poorter et al. [3] demonstrated that indoor experiments often use low levels of light, which might reduce plant biomass in comparison with outdoor-grown plants.

While the effect of light quality on the aboveground organs was quite similar among species in the current study, the direction of the effect on roots was clearly species dependent. With species such as *Alnus* and *Ocimum* exhibiting higher root growth at very low and high B%, and species such as *Raphanus* and *Ulmus* showing increased root production at intermediate B percentages (25 and 35% B). To date, scarce information is available on the effects of light quality on belowground plant productivity. A previous study by Yorio et al. [28] reported that under 10% B mixed with 90% R light there was a higher root production in Lactuca, Raphanus, and Spinacia, compared with plants grown under pure R light. Nhut et al. [29] found that mixtures of B and R light stimulate the production of roots compared with pure R light in strawberry plantlets. Independent of light quality, we found a significantly enhanced root production in the phytotron treatments compared to the field grown plants, except for the 62% B treatment. As indicated by Poorter et al. [3], indoor climatization might induce root zone conditions that differ markedly from field conditions, leading to altered root production and consequently profoundly changed plant growth. As all plants in our experiment were regularly watered in both field and phytotron treatments, we can exclude that the observed higher root productivity in the phytotrons results from different water availability between indoor and field trials. However, pot soil temperature was not monitored, and it is possible that it differed significantly between indoor and field conditions, partly due to the lack of infrared radiation from the LED lamps.

### 3.2. Light Quality Effect on Leaf Pigmentation

The concentration of chlorophyll and carotenoids changed strongly with light quality in our study. Under natural sunlight, cryptochrome activity is reduced at high radiation, thereby signalling strong light conditions in the plant. The same effect can be achieved under experimental conditions by exposing plants to high percentages of B light [30]. The high proportion of B light in our 62% B treatment thus triggered the enhanced production of photosynthetic pigments despite the fact that the other treatments with lower B% had the same PPFD. In fact, the low concentrations of Chl a and b in plants that have been treated with low levels of B light or monochromatic R light in previous studies, have even led to photo-oxidative stress in plants due to an increase of O_2_^-^ and H_2_O_2_ radicals that induce cellular damage [8,19]. Barnes and Bugbee [30] proposed that a minimum of 20−30 μmol m^−2^ s^−1^ of B light is necessary to reach natural-like growth and morphologies, even if such a minimum requirement for B light appears to be highly species-specific [31]. It is likely that due to all of our light treatments including at least 6% of B light, we did not observe light quality related stress effects in our experiment. However, we identify that even with over 30 μmol m^−2^ s^−1^ of B light (at 6% B), higher percentages of B can increase the photosynthetic maximum capacity in several species, indicating that it is not just the quantity of B light, but also its relationship with other wavebands in the spectrum. Interestingly, most species showed higher Chl a:b ratios in the phytotrons compared to the field trial. This effect has been observed previously in indoor-grown plants [32], where it is attributed to the lack of fluctuating light conditions in indoor facilities.

Like chlorophyll, the production of carotenoids was also significantly increased with 62% of B light compared to 6% B (and 35% B), yet only the 25% B and the 62% B treatments induced higher carotenoid concentrations than in the field trial. Hogewoning et al. [8] reported an increase of carotenoids in cucumber plants when B was increased to 50% in the light spectra. An increase of carotenoids has been shown to work as an accumulative protection mechanism correlating with high light intensities or high B ratios. For example, the authors of [12] found that Fv/Fm of rapeseed leaves was reduced under monochromatic B or R light treatments, compared with mixtures of B and R. They attributed this to a higher PS II damage and linked the higher concentrations of carotenoids to a protection mechanism against oxygen radical formation. This is in line with our Fv/Fm results, where lower percentages of B in the applied spectra induce small but significant differences of the Fv/Fm values in almost all investigated species.

### 3.3. Light Quality Effects on Photosynthesis

When A_max_ was measured under the same standardised light conditions (30% B and 70% R) in the current study, plants under 63% B showed, on average, significantly higher A_max_ compared to plants under 25% B and the field trial. This could be partially explained by the increased chlorophyll concentrations in 63% B treated plants (see above). Previously, higher A_max_ have been linked to higher levels of stomatal conductance and nitrogen concentration, where the latter is correlated to Rubisco, cytochrome, proteins and chlorophyll content [33]. A higher A_max_ has also been suggested to partially derive from an instantaneous stimulation of photosynthesis (i.e., during the exposure to the light within the gas-exchange chamber) due to the lack of adaptation to the standardised light condition [8]. In our case, using 70% R in plants adapted to 62% B may promote a higher A_max_, meanwhile this may not be the case in plants adapted to lower percentages of B light, and therefore higher percentages of R light. Kim et al. [15] have shown that in Pisum sativum about four days were necessary to reach full photosynthetic acclimation after a transition from a PSI to a PSII stimulating light environment and vice versa. Similarly, Hogewoning et al. [34] showed in duckweed, that six days were needed to fully acclimate to different light conditions, using the Chl a:b ratio as the control parameter.

In contrast to the measurements of standardised light, when measured under the respective in situ light conditions, A_max_ was significantly lower at very low (6%) or very high (62%) B light conditions, despite the higher concentration of chlorophyll at 62% B or small differences in SLA (Figure 2B). In a similar but more extreme experiment, several long-term studies reported lower net photosynthesis or A_max_ in plants raised under monochromatic B or R light [8,11,12]. Hogewoning et al. [8], also reported dysfunctional photosynthesis in cucumber plants, grown under pure R light and a dose response curve in A_max_ when the B% was increased up to 50% B, with no further increase of A_max_ beyond 50% B. The increase of A_max_ with B percentages was associated with a reduction of the SLA, an increase of N and chlorophyll per leaf area, and higher stomatal conductance under mixtures of B and R light compared with only B or R [8]. Matsuda et al. [35] reported an increase of A_max_ in spinach plants exposed to a 1:1 B: R radiation compared with just B light, associated with increased leaf N concentration. Shengxin et al. [12] showed that dark adapted Fv/Fm values were higher (as an indicator for less photo-stress) under mixtures of B and R light compared with monochromatic B or R light.

The effects of treatments on photosynthesis were also visible in the quantum yield of the CO_2_ fixation curve (α) of the investigated species. Similar to A_max_, a more natural level of B light may explain a higher efficiency when an ‘in situ’ light was used for our gas-exchange measurements, with significantly higher values indoor than in the field trial. Similar results have been reported at 15–30% B compared with 50% B [8]. This effect may indicate the evolutionary adaption of species to the natural sunlight spectrum, with higher quantum yield under a more natural B:R ratio (circa 33% of B in the sunlight spectrum [36]). Other conditions with extreme levels of B or R light may require the adaptation to each light condition, where CO_2_ fixation may have a wavelength dependence related to absorption properties of the different pigments involved. Terashima et al. [37], described three major causes for the wavelength dependency of the quantum yield: absorption by photosynthetic carotenoids, absorption by non-photosynthetic pigments and an imbalanced excitation of the two photosystems, where an imbalance in excitation will result in quantum yield losses [27,38]. It has been shown that a correct light stimulus, with light qualities matching the species-specific ratio of PSII and PSI, is key to high quantum efficiency of photosynthesis [39]. The light compensation point of photosynthesis (CP) was generally not affected by light quality. Similar results have been observed in previous cases [9,12].

In the current study, the average dark respiration (DR) using the standardised light, independent of the species, was relatively lower at 62% B compared with the other light treatments or the field trial. Atkin et al. [40] described in tobacco that observed changes in DR were dependent on the previously applied irradiance (tested between 0 to 300 μmol photons m^−2^ s^−1^). An instantaneous stimulation of the photosystems in low light adapted plants due the stimulus of an intensity radiation burst was hypothesised. Although the total photon flux was the same between treatments in our study, similar short time effects on DR might have occurred when plants were exposed to a high intensities and light spectrum that they were not adapted to.

### 3.4. Principal Component Analysis

The PCA analyses performed in this study confirmed that the effects of light quality on plant performance are highly species dependent, and adjustments of the light spectra may help to promote more natural like growth, where more natural growth like plants tend to group closer to the field trial in the PCA. Applying a light spectrum with similar B and R light proportions to sunlight is proposed to avoid physiological plant responses to a lack or excess of B light (which might also differ among species). Although 7% B has been recommended to avoid dysfunctional photosynthesis [8], this study indicates that levels of 25 to 35% B light in the spectrum are needed in indoor conditions to avoid undesired (i.e., unnatural) effects of the light spectrum on plant growth. This was demonstrated with higher distances of the 6%B light treated plants from the field trial plants in the PCA. No specific trait was identified across the different species to have a higher importance than others (Appendix A), where the ranking of importance of each measured parameter was species dependent. Independent of this, the PCA clearly indicated that other environmental variables should be controlled (e.g., air flux, soil temperature) or more precisely mimicked in indoor growth facilities if natural-like growth is required. A similar approach has been previously used [41] to understand the difference between indoor and outdoor experiments, with a focus on *Arabidopsis*’s metabolism where a clearer clustering of the indoor and outdoor conditions was obtained. Similar values of the first and second component to the ones presented here (first and second component explaining 28 and 15% of the variance, respectively compared with 24 and 15% average across species in our study).

## 4. Materials and Methods

### 4.1. Plant Material and Pre-Growing Conditions

In this study, we investigated young plants of 7 species from different functional plant types to include the species as the source of variation: trees represented by black alder (*Alnus glutinosa* (L.) Gearth, provenance HG4, Zurich, Switzerland), Scotch elm (*Ulmus glabra* Huds., provenance Merenschwand, Aargau, Switzerland), herbs represented by basil (*Ocimum basilicum* ‘Adriana’), lettuce (*Lactuca sativa*), melissa (*Melissa officinalis*), radish (*Raphanus raphanistrum* subsp. sativus (L.) Domin), and grasses represented by winter wheat (*Triticum aestivum*). For the experiments, all plants were raised from seeds. The seeds of both tree species were purchased from the Swiss federal institute for forest, snow and landscape research, WSL, Birmensdorf, Switzerland. All herb seeds were provided from Wyss Samen und Pflanzen AG, Zuchwil, Switzerland, and *Triticum* seeds were supplied form Sativa AG, Rheinau, Switzerland. Hereinafter, the species will be referred to by their scientific genus name for clearness. Due to the different germination speeds the timing of sowing was different for the species as follows: seeds of *Alnus* and *Ulmus* were sown in 20 × 40 × 2 cm trays with commercial substrate (pH 5.8, 250 mg L^−1^ N, 180 P_2_O_5_ mg L^−1^, K_2_O 480 mg L^−1^, Ökohum, Herrenhof, Switzerland) 43 days before the start of the experiments and were left to germinate under 190 μmols m^−2^ s^−1^ of photosynthetic photon flux density (PPFD: 400–700 nm) with 25% Blue (B: 400–500 nm), 32% Green (G: 500–600 nm) and 41% Red (R: 600–700 nm) light and an R to far red (FR: 700–800 nm) ratio (R:FR. 655–665 nm and 725–735 nm; according to [42]) of 5.1 for 23 days, using LED lighting with a day length of 16 h. Twenty days before the start of the experiment, the light was increased to 240 μmols m^−2^ s^−1^ PPFD, with a R: FR of 5.1, to acclimate the plants to higher intensity levels. Thirteen days before the start of the experiment *Melissa* seeds were sown in the same type of trays and keeping the last-mentioned environmental conditions. Six days before the start of the experiments the remaining species were sown in the same type of trays and under the same environmental conditions, with the exception of *Triticum,* which was sown immediately in round 2 L pots with a density of 15 seeds per pot (13.5 cm diameter, Poppelmann, Lohne, Germany). All light measurements were done using a using a spectrometer (STS, OceanOptics, Florida, United States). During the germination and the pre-treatment period, the different seedlings were raised at 25 °C/50% relative humidity (RH) during daytime and 15 °C/83% RH during night, with 10 h per day and one-hour light/temperature/humidity ramping pre and post day.

At the start of the experiment, all species, excluding *Triticum*, were transplanted to the same type of 2 L pot previously used for Triticum, with a single individual in each pot. Moreover, Triticum was thinned to 10 plants per pot. The pots were filled with the same substrate as used in the germination trays, and 4 g of Osmocote slow release fertiliser (Osmocote exact standard 3–4, Scotts, Marysville, OH, USA), containing 16% total N, 9% P2O5, 12% K2O and 2.5% MgO, was added to each plot. All plants were watered daily in the morning throughout the experiment.

The pre-growing procedure was repeated 3 times for this study: First, for the field-trial that was used as reference for the phytotron experiments, and then twice for the different light treatments of the phytotron experiment. (See control and light quality treatments below). No significant difference in initial height or biomass was found at the start of the experiments within species for the different replications (data not shown).

### 4.2. Control and Light Quality Treatments

To establish a control treatment as a reference point for natural growth, all seven target species were grown in a field trial for 35 days (4 August 2017–7 September 2017) at the botanical garden of the University of Basel, Switzerland. Throughout the field trial, the in situ climate and the natural sunlight spectrum was recorded (Appendix A and below). Following the field trial, we exposed plants from the seven different species to four mixtures of B and R light, which can be expressed as a B/R ratio, or as percentage of B light in four walk-in Phytotrons (1.5 m × 2.5 m) with full control of temperature, air humidity and light quality and quantity (prototypes, Enersign GmbH, Basel, Switzerland). To unify nomenclature with previous studies, the four different light treatments will be referred to by their respective B light proportion (Table 1). The light treatments were chosen based on previous literature (e.g., Hogewoning et al. [8]), measurements of natural light completed in situ [36], and technical capacities of the phytotrons at the average light intensity of the outdoor treatment. For each treatment, the replication per species was 9 pots (with either one or more individuals per pot depending on species; see above). In all light treatments, the average PPFD from the field trial (575 μmol m^−2^ s^−1^) was provided at the average height of the different species using 18 LED panels for each chamber consisting of a mixture of B (400–500 nm)**,** White (2500 K), R (600–700 nm) and FR (700–800 nm) LEDs per panel (prototypes, DHL-Licht, Hanover, Germany). The LED lighting system of each chamber was mounted on movable ceilings, the height of which can be adjusted through the environmental control software of the chambers. To preserve similar light levels at average plant height, the height of the lamps was adjusted twice during the experiment. Based on the field trial conditions, the day length was set to 13 h and 5 min, giving a constant daily light integral (DLI) of 27.1 mol m^−2^ day^−1^ in all light treatments. Similar to the light conditions, temperature and humidity during day and night were set to average field trial conditions: 22 °C/66% RH and 18 °C/79% RH, for day and night, respectively, with a period of one-hour ramping before and after daytime. A uniform temperature and humidity distribution within each chamber was ensured by a constant vertical air stream from below. To avoid border and space effects, all plants were randomly distributed within each phytotron on two tables. The tables were rotated by 90° every day. Each light treatment was replicated twice (two separate runs of all four light combinations), where the distribution of the chambers was random between the two runs.

At the end of the 35-day experimental period, a suite of measurements was conducted in the field trial and the phytotron experiments. A description of the measured parameters is given in the following paragraphs. Due to limitations imposed by the lamp characteristics at high intensities, a higher R:FR ratio compared with outdoor (1.8 vs. 1.1) was applied in order to reach the targeted light intensities. No UV light was applied in the phytotrons.

### 4.3. Climatic Growth Conditions

In order to apply the most natural conditions within the phytotrons, the climate from the field trial at the botanical garden of the University of Basel, Switzerland, was recorded throughout the 35-day growth period (Appendix A). Relative humidity, temperature, and PPFD were measured every 5 min with a weather station (Vantage pro2, Davis, Haywards, CA, USA). In addition, sunlight spectra in the waveband 350–800 nm were recorded every minute using a spectrometer (STS) that was equipped with an optical fiber and a cosine corrector (180º field-of-view; CC-3-UV-S, OceanOptics) placed by the weather station’s PAR sensor facing upwards. The spectrometer was connected to a Raspberry Pi 2 computer for automatic sampling, integration time adjustments and data storage. A posteriori, the spectra were used to calculate photon flux densities within specific wavebands: PAR, B, G, R and FR. The PAR light measurements were verified by comparing the data from the weather station with the data from the spectrometer readings. The data from the field trial were used to calculate average diurnal and nocturnal temperature, air humidity and PAR conditions for the phytotron treatments.

### 4.4. Morphological Parameters

By the end of the 35-day growth period, plant height was measured as total height from the substrate to the apical tip. In the case of long inflorescences (*Raphanus*) or plants without a clear stem (*Triticum*), extended leaf length was recorded as height, and in the case of *Lactuca*, no height was recorded. Two full-grown leaves from the top three mature leaves were collected from each plant to measure leaf area (LI-3100, Licor, Lincoln, NE, USA) and calculate the specific leaf area (SLA) in cm^2^ g^−1^ on a dry leaf weight basis. Dry weight (DW) was measured separately for leaves, stems and roots after 10 days drying at 80ºC in a drying oven (UF 260, Memmert, Schwabach, Germany). Due to the lack of a clear stem, only total aboveground and root biomass were measured for *Lactuca*, *Melissa* and *Triticum*. All reported organ weights and the below to above ground biomass ratio (root:shoot-ratio) refer to plant dry mass.

### 4.5. Chlorophyll Fluorescence and Chlorophyll Content

One night before the end of the experiment, fast chlorophyll fluorescence induction was measured on one of the top three leaves in four randomly chosen plants of each species and treatment by using a continuous excitation fluorometer with an intensity of 3500 μmol m^−2^ s^−1^ centred at 627 nm (Pocket PEA, Hansatech instruments Ltd., Norfolk, UK). The plants were dark adapted for at least 20 min before recording photosynthetic maximum quantum yield (Fv/Fm) and the absolute performance index (PI) of the leaves, which has been correlated previously to stress (for calculations and details, see [43]).

During harvest, two discs of 1.13 cm^2^ area from the top four leaves were punched and stored in a 1.5 mL Eppendorf tube together with four to six glass beads of 0.1 mm diameter for later chlorophyll analysis. The tubes were quickly frozen in liquid nitrogen and then kept at −80 °C until analysis. During the day of chlorophyll measurement, the tubes were agitated two times for 10 s to triturate the tissue using a mixing device (Silamat S6, Ivoclar Vivadent, Schaan, Liechtenstein). After adding 0.7 mL of acetone to each tube, they were agitated again for 10 s and then centrifuged at 13,000 rpm at 4ºC for 2 min. A total of 0.25 mL of the supernatant was dissolved in 0.75 mL of acetone, and the sample absorption spectra were measured using a spectrometer (Ultrospec 2100 pro, Biochrom, Holliston, MA, USA). Chlorophyll a and b concentrations, chlorophyll a to b ratio (Chl a, Chl b and a:b ratio, respectively) and total carotenoid concentrations as mg g^−1^, were calculated from the spectra using the values at 470, 646 and 663 nm as described in [44].

### 4.6. Leaf Gas Exchange

Six days before the end of the experiment, a light response curve of net CO_2_ leaf-exchange was measured in one of the top three leaves in three randomly chosen plants per species and treatment using a LI-6800 photosynthesis system (LI-COR, Lincoln, NE, USA). The light response curves were measured under two different light spectra: (i) a standardised artificial light spectrum, composed of 70% R and 30% B (in the following referred to as ‘standardised light’) provided by the chamber head light source to study photosynthesis of the different species under a uniform light spectrum, and, (ii) the respective growing light spectrum (in the following referred to as ‘in situ spectrum’) provided by using a transparent, clear-top chamber head (Clear-top leaf chamber 6800-12A, LI-COR) to study photosynthesis of the different species under their respective growing spectra and avoid any bias on photosynthesis from a non-adapted spectrum. Twelve different light intensities: 2000, 1500, 1000, 800, 600, 400, 200, 100, 50, 25, 10 and 0 μmol m^−2^ s^−1^ of PPFD were used for light response curves with the ‘standardised light’ spectrum. Due to lower maximum irradiance in the phytotrons limited by the light quality being applied (see above), the light response curves for the ‘in situ’ growing light were measured only up to a maximum radiation of 700 μmol m^−2^ s^−1^ of PPFD (700, 480, 380, 200, 100, 60, 30, 20, 17, 15 and 0 μmol m^−2^ s^−1^ of PPFD). All leaf CO_2_-exchange measurements were conducted at 400 ppm CO_2_, 60% relative air humidity and 20 °C leaf temperature, with 60 to 120 s as the threshold for stability after each light change intensity. Stability of readings was assumed when the difference of the slopes between IRGA’s were smaller than 0.5 μmol mol^−1^ sec^−1^ and 1 for CO_2_ and H_2_O, respectively.

For each light curve, 12 different light models were fitted accordingly [45], including a model for photo-inhibition [46]. For each species and treatment, the model with the best fit (lowest sum of squares) was selected (details in [45]). The selected model was then used to calculate the following four values from the light response curve: maximum photosynthesis within the range of measured light (A_max_), quantum yield of the CO_2_ fixation (α) as the slope of the linear curve between 0 and 100 μmol m^−2^ s^−1^ of PPFD, dark respiration (DR) and the light compensation point (CP) of photosynthesis.

### 4.7. Statistical Analysis

To evaluate the effect of the light treatments, a two-way analysis of variance (ANOVA) was performed for all measured parameters, considering the species and different treatments as fixed factors and the two replicates of each treatment as a random factor. The significance of the random factor was evaluated using a restricted likelihood ratio test. The data were checked for normal distribution, independence and homogeneity of the variance.

To enable the direct visible and statistical comparison of the treatment effects across species, each measured trait was normalised relative to its mean value on the field trial for each species (the original trait average values per species and treatments are available in Appendix A). The normalised values were used to perform a one-way ANOVA, considering the treatments as fix factor and species as random factors (Appendix A). A Tukey pairwise multiple comparison test was used as post hoc analysis to identify significant differences (*p* < 0.05) among treatments. In several cases when all indoor light treatments differed from the field trial, an additional one-way ANOVA was performed without the field trial to highlight the individual response differences to the different light treatments (Data non shown).

Finally, to identify the specific traits that have the maximum variation between treatments and to quantify which treatment gave the overall most similar response compared to the outdoor trial, a principal component analysis (PCA) was performed separately for each species, using the different measured traits as input values. To perform a PCA analysis, the same number of observations is required for each variable but due to fewer photosynthesis measurements, chlorophyll measurements and fluorescence measurements than the number of plants used for biomass measurements, in each species and treatment, the missing values of chlorophyll content and light parameters were imputed using normal distribution with the same average and standard deviation of the available data. All analyses were performed using R [47] and the package plyr for data processing and lm4, car, RLRsim, emmeans for data analysis and multicomp and vegan for statistically significant representations.

## 5. Conclusions

The applied light spectra in this study significantly influenced plant morphology, pigment concentration and photosynthesis. Less deviating responses compared to the field trial were reached with either 25% or 35% of B light in almost all species. Hence, if natural like plant growth is desired in indoor plant cultivation, the application of a balanced light spectrum is generally recommended. Despite this, spectral quality of the light source is only one of many factors that can potentially bias plant performance. In this study, we thus aimed to apply similar climatic conditions within the growth chambers as were measured in the field trial to compare outdoor with indoor growth. Nevertheless, we still found significant differences between phytotron and field grown plants in most of the investigated plant traits. This highlights the difficulties to exactly reproduce natural plant performance in indoor growth facilities, as well as the necessity to include the simulation of additional environmental factors (e.g., replication of natural minimum and maximum temperature, humidity and irradiance changes, wind speed and direction) in indoor experiments with plants.

## Figures and Tables

**Figure 1 plants-09-01273-f001:**
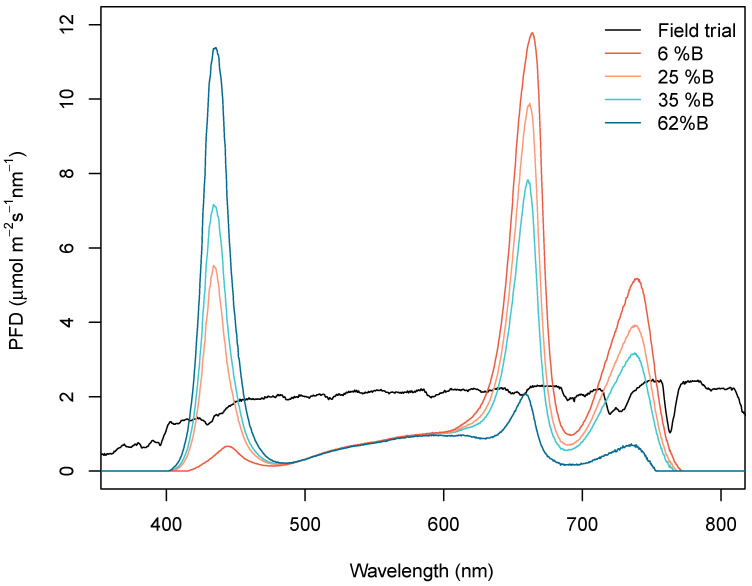
Applied spectra for the field trial and each of the different light treatment where 6%, 25%, 35% and 62% refers to the percentage of blue light as percentage of the photosynthetic photon flux density (PPFD) (In other words, excluding far-red). The integrated area between 400 and 700 nm corresponds to an approximate 575 μmol m^−2^ s^−1^ of photosynthetic photon flux density in each case.

**Figure 2 plants-09-01273-f002:**
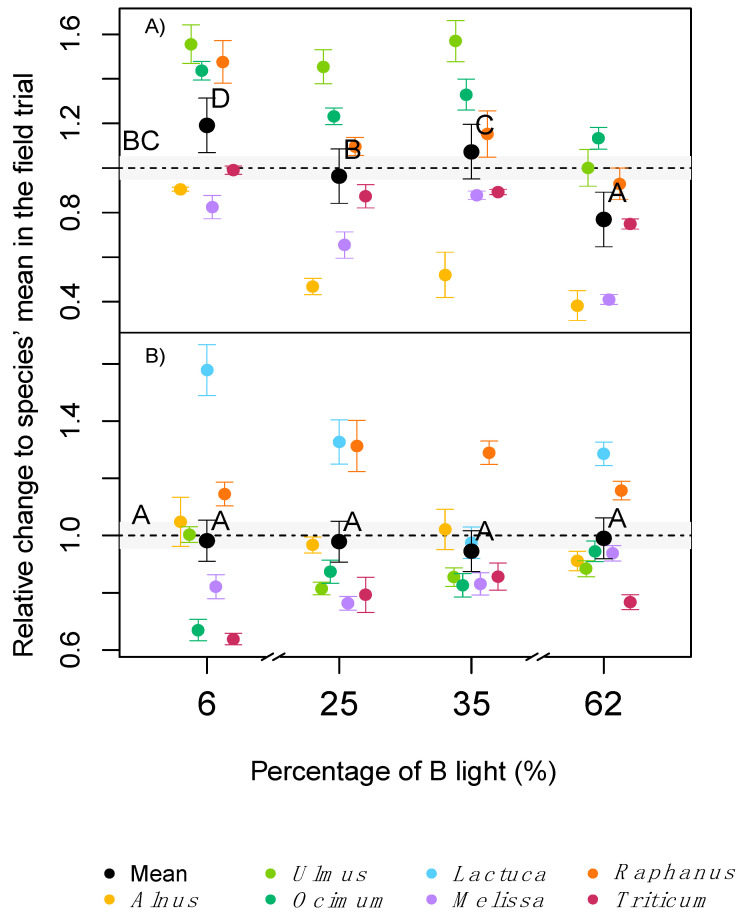
Fold change on: plant height (**A**) and SLA (**B**), relative to the average field trial (dotted line). Coloured dots are the average of each species in both experiment runs (*n* = 18), the black dots are the average values across all 7 species (*n* = 126). Error bars indicate the standard errors. The grey area corresponds to the standard error of the field trial. Different letters indicate statistical difference between groups with experiment replicate and species as a random effect.

**Figure 3 plants-09-01273-f003:**
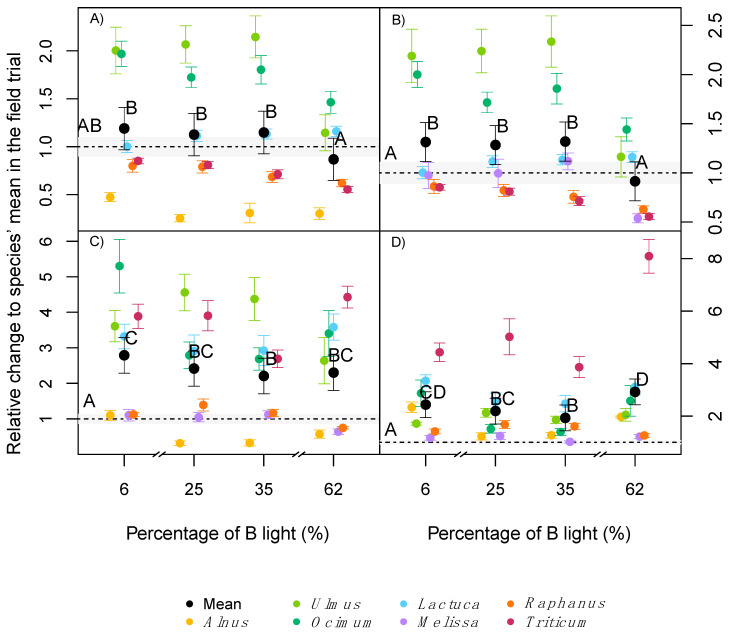
Fold change on: leaves (**A**), shoot (**B**), roots (**C**) and root to shoot ratio (**D**), as dry weight relative to the average value of the field trial (dotted line). Coloured dots are the average of each species in both experiments runs (*n* = 18), the black dots are the average values across all 7 species (*n* = 126). Error bars indicate the standard errors. The grey area corresponds to the standard error of the field trial. Different letters indicate statistical difference between groups with experiment replicate and species as a random effect.

**Figure 4 plants-09-01273-f004:**
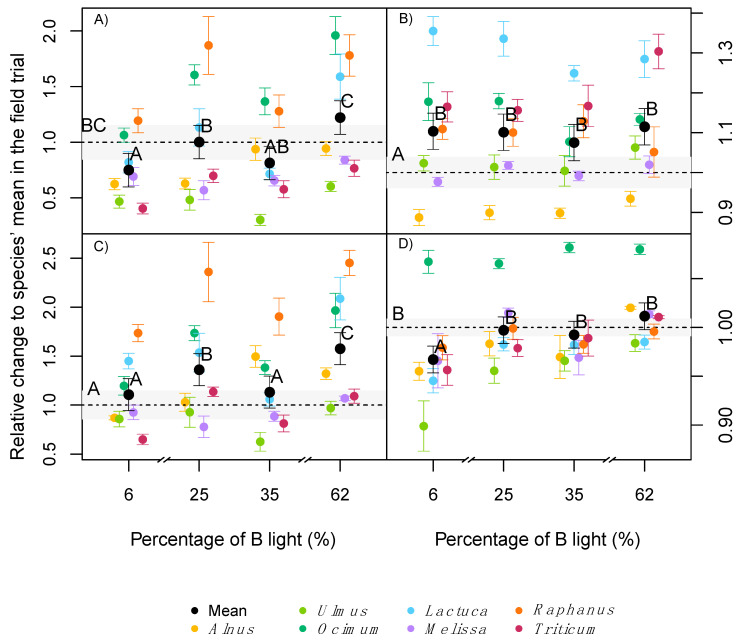
Fold change on Chlorophyll a (**A**), Chlorophyll a:b ratio (**B**), carotenoids content (**C**) and Fv/Fm values (**D**) relative to the average value of the field trial (dotted line). Coloured dots are the average of each species in both experiments runs (*n* = 18), the black dots are the average values across all 7 species (*n* = 126). Error bars indicate the standard errors. The grey area corresponds to the standard error of the field trial. Different letters indicate statistically difference between groups with experiment replicate and species as a random effect.

**Figure 5 plants-09-01273-f005:**
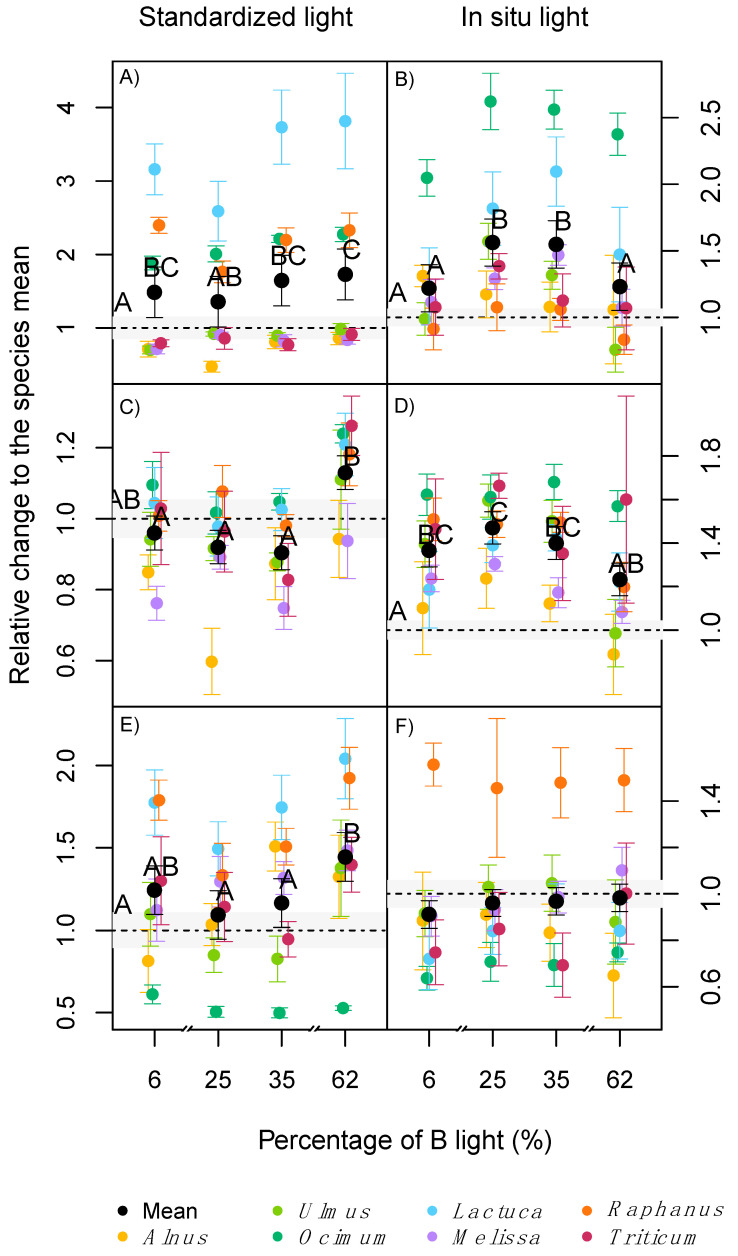
Fold change on maximum photosynthesis (*A_max_*, **A**,**B**), quantum yield of the CO_2_ fixation curve (α, **C**,**D**) and dark respiration (DR, **E**,**F**) relative to the average value of the field trial (dotted line). Values were measured with either a standard light with 70% B light and 30% R light (‘standardised light’) or the actual ‘in situ’ light (see methods for details). Coloured dots are the average of each species in both experiments runs (*n* = 18), the black dots are the average values across all 7 species (*n* = 126). Error bars indicate the standard errors. The grey area corresponds to the standard error of the field trial. Different letters indicate statistical difference between groups with experiment replicate and species as a random effect.

**Figure 6 plants-09-01273-f006:**
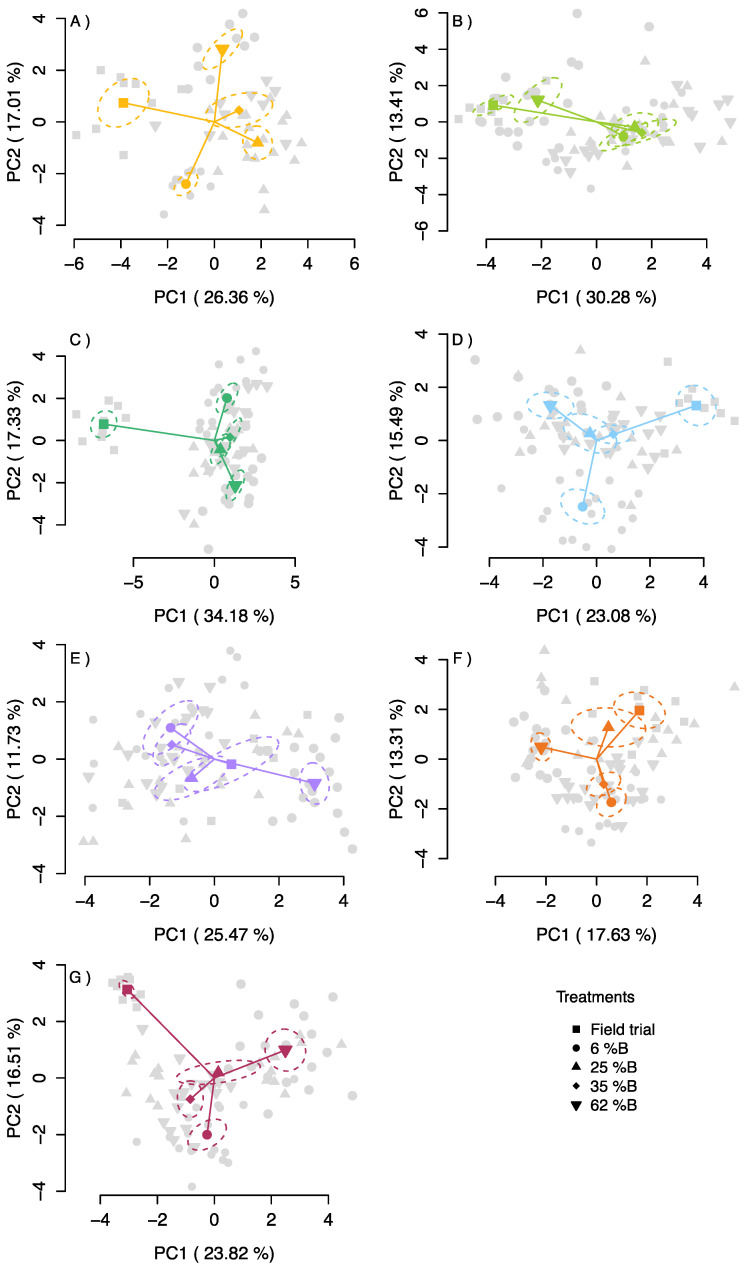
Principal component analysis (PCA) of the measured traits of each species: (**A**) *Alnus*, (**B**) *Ulmus*, (**C**) *Ocimum*, (**D**) *Lactuca*, (**E**) *Melissa*, (**F**) *Raphanus* and (**G**) *Triticum*, grown under 6% B, 25% B, 35% B and 62% B light. Each lighter point (*n* = 18) corresponds to a plant and solid ones to the average weighted centroids of each light treatment, where the name of each species is mentioned in the respective upper right corner. Ellipses correspond to the standard error of the weighted centroids with a confidence interval of 95%.

**Table 1 plants-09-01273-t001:** Spectral characteristics of sunlight and of the indoor light treatments, based on the measured spectra shown in Figure 1.

Treatment\Characteristic	Field Trial	6% B	25% B	35% B	62% B
Blue (%)	28	6	25	35	62
Green (%)	36	16	16	16	16
Red (%)	36	78	59	49	22
R:FR ratio	1.1	1.8	1.8	1.8	1.8

**Table 2 plants-09-01273-t002:** *p*-values derived from the full-factorial ANOVA analyses of the different measured plant traits, with light treatment and species as fixed factors, and the replicates of the individual light treatments as random factors. Non-significant *p*-values (≥0.05) are indicated as “-”.

Type of Factor	Fix Factors			Random Factors
**Factor**	**Light Quality**	**Species**	**Light Quality × Specie**	**Replicate**
**Variable**				
**Biomass and Morphology**				
Height *	<2.2 × 10^−16^	<2.2 × 10^−16^	<2.2 × 10^−16^	5 × 10^−4^
Dry weight leaves	1.16 × 10^−5^	<2.2 × 10^−16^	<2.2 × 10^−16^	1.5 × 10^−3^
Dry weight shoot **	1.03 × 10^−8^	<2.2 × 10^−16^	<2.37 × 10^−14^	-
Dry weight roots	1.26 × 10^−5^	<2.2 × 10^−16^	<2.2 × 10^−16^	<2.2 × 10^−16^
Total dry weight	8.74 × 10^−5^	<2.2 × 10^−16^	<2.2 × 10^−16^	<2.2 × 10^−16^
Root to Shoot ratio	1.39 × 10^−11^	<2.2 × 10^−16^	<2.2 × 10^−16^	<2.2 × 10^−16^
SLA	0.1024	<2.2 × 10^−16^	<2.2 × 10^−16^	7.9 × 10^−3^
**Chlorophyll**				
Chlorophyll a (mg g^−1^)	4.90 × 10^−7^	<2.2 × 10^−16^	3.47 × 10^−14^	<2.2 × 10^−16^
Chlorophyll b (mg g^−1^)	<2.2 × 10^−16^	<2.2 × 10^−16^	<2.2 × 10^−16^	5.62 × 10^−14^
Chl a:b ratio **	1.85 × 10^−5^	<2.2 × 10^−16^	5.98 × 10^−6^	-
Carotenoids (mg g^−1^)	1.49 × 10^−13^	<2.2 × 10^−16^	2.78 × 10^−13^	<2.2 × 10^−16^
Fv/Fm **	2.53 × 10^−8^	<2.2 × 10^−16^	0.003297	−
**Standardised light**				
Max photosynthesis **	0.03074	4.42 × 10^−5^	3.09 × 10^−8^	-
Quantum yield **	2.44 × 10^−6^	1.94 × 10^−12^	-	-
Dark respiration **	0.4026571	9.16 × 10^−12^	6.89 × 10^−5^	−
Compensation point	0.008619	<2.2 × 10^−16^	5.48 × 10^−11^	<2.2 × 10^−16^
**In−situ’ light**				
Max photosynthesis **	6.52 × 10^−6^	1.25 × 10^−12^	-	-
Quantum yield **	6.45 × 10^−6^	1.93 × 10^−7^	-	-
Dark respiration **	-	4.06 × 10^−6^	-	-
Compensation point	0.3041	4.19 × 10^−16^	1.74 × 10^−5^	<2.2 × 10^−16^

* Lettuce was removed from these analyses. ** Interactions or factors were removed from the analysis due non-significance.

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
