# Peer review of "Reaching Natural Growth: Light Quality Effects on Plant Performance in Indoor Growth Facilities"

_plants, 2020, doi:10.3390/plants9101273_

Round 1

Reviewer 1 Report

It is a interesting paper, evaluating the effects of varying light spectra on indoor growth of differenti functional plant types. Differently from the majority of the other studies, in this research, all the the environmental  parameters are mantained the closer possibile to the outdoor conditions.A few suggestions  to improve manuscript are thereafter suggested:

Sub-paragraph 2.1 of the results section should be inserted in the Materials and Methods one.

Row 419: Do "the same environmental conditions" mean 240 micro mole m-1  s-1 or the initial light conditions? Specify

Row 433:eliminate the point after 'experiment'

Row 489: correct "cm2 g-1

Row 507: eliminate the point after mL

Row 512: correct 'cm2'

Figure 6:  it is useful to repeat the legend with the colours of each species under the figure.

Due to the novelties of the results, to the  appropriate methodology . I think this paper may be' accepted wiith minor revisions.

Author Response

Dear reviewer.
 We apreciate your review and the changes that you suggested were incorporporated.

Thanks for your support

The authors

Reviewer 2 Report

The manuscript describes the indoor plants growth affected by the combination of the Blue and Red LED light. Several published papers have discussed the LED light quality that has been proved an essential roles on the plan growth. However, the current manuscript still proves useful results for the interested readers. Several suggestions:

  1. The whole manuscript has to be examined by the English native speakers. For example, in the introduction part, page 72, "The aim of this study, as a first step on a series of experiments to reach natural-like growth under indoor conditions, was to investigate the effects of varying proportions of B and R light within walk in growth chambers (phytotrons) on growth and physiological traits of plants from different functional plant groups." This sentence should be separated into two sentences to clearly show the exact meaning that the authors tried to describe. Also "The inclusion of seven different species form different functional plant types...." there should be "and" between form and different.....
  2.  What's the definition of "Far Red" light? Infrared light? 
  3. It seems like the current manuscript is an continuous studies from previous published paper, the current manuscript should describe some results from previous studies. For example, in the end of introduction, "In contrast to many previous studies, we explicitly applied more natural-like R:FR ratios and light intensities, and the plants were exposed to temperature and air humidity based on the pre-measured field trial.", What's the previous studies? What's the results from pre-measured field trail? At lease, the manuscript should put the references.
  4. Table I, R:FR ratio should be 1:1 not 1.1
  5. Since the MDPI support color figures, The curves in Figure 1 should be changed to color curves.
  6. Generally, it is not easy to tell the plant growth from the PCA analysis. The manuscript should use a table to clearly show the results from PCA analysis.
  7. Why Figure 5 has different size comparing to Figure 3 and 4?
  8. Materials and Method section should be move to section 5 instate of section 4. 

Author Response

Dear reviewer

We apreciate your review and in base to your review we would like to comment:

The manuscript has been proofreaded by a native english speaker and all your specific requierements has been added (Definition of far red light, missing references and color figures), including a more clear discussion of the PCA.

Regarding the R:FR ratio we would like to comment that was 1.1 and not a typo. Similar, the figures had a different sizing for presentation reasons. Finally the conclusion is presented as point number five, following the template of the journal plants.

We hope that this changes satisfies your requierements and we would like to thanks for your help,

The authors

Round 2

Reviewer 2 Report

The revised manuscript has all answered reviewer's comment, it could be accepted for publication.

Author Response

Thanks for your report